# High-throughput robust single-cell DNA methylation profiling with sciMETv2

Ruth V. Nichols [1], Brendan L. O'Connell [1,2], Ryan M. Mulqueen [2], Jerushah Thomas[3], Ashley R. Woodfin[3], Sonia Acharya[1], Gail Mandel [4], Dmitry Pokholok[3], Frank J. Steemers[3] & Andrew C. Adey [1,2,5,6] ✉

DNA methylation is a key epigenetic property that drives gene regulatory programs in development and disease. Current single-cell methods that produce high quality methylomes are expensive and low throughput without the aid of extensive automation. We previously described a proof-of-principle technique that enabled high cell throughput; however, it produced only low-coverage profiles and was a difficult protocol that required custom sequencing primers and recipes and frequently produced libraries with excessive adapter contamination. Here, we describe a greatly improved version that generates high-coverage profiles (~15-fold increase) using a robust protocol that does not require custom sequencing capabilities, includes multiple stopping points, and exhibits minimal adapter contamination. We demonstrate two versions of sciMETv2 on primary human cortex, a high coverage and rapid version, identifying distinct cell types using CH methylation patterns. These datasets are able to be directly integrated with one another as well as with existing snmC-seq2 datasets with little discernible bias. Finally, we demonstrate the ability to determine cell types using CG methylation alone, which is the dominant context for DNA methylation in most cell types other than neurons and the most applicable analysis outside of brain tissue.

Much like other genomic properties, the benefits of single-cell DNA methylation assays are substantial: providing the ability to identify cell types and states within a complex tissue[1]. Currently the primary means of assessing DNA methylation is using strategies to convert non-methylated cytosine bases into uracil, which are in turn sequenced as thymine, whether chemical in the form of sodium bisulfite, or enzymatic. This paradigm requires a multi-stage process with buffer exchanges and cleanups, imposing a significant hurdle for high-throughput single-cell assessment. As such, nearly every single-cell DNA methylation assay begins with isolating individual cells or nuclei into individual reaction wells for bisulfite or enzymatic conversion and

subsequent processing; imposing significant challenges for both cost and scaling[2–8].

To address this, we previously described a single-cell combinatorial indexed assay for the assessment of DNA methylation (sciMET)[9]. This technique enables the encapsulation of genomic DNA within intact nuclei during an initial round of cell barcoding. Subsequent processing, including bisulfite conversion, adapter tagging, and PCR amplification is then carried out on pools of cells; greatly reducing costs and enabling high throughput. To accomplish this, sciMET relies on the fixation of nuclei followed by the disruption of nucleosomes, as first described in sci-WGS for single-cell whole genome sequencing using combinatorial indexing[10]. This process enables uniform access to

[1]Department of Molecular & Medical Genetics, Oregon Health & Science University, Portland, OR, USA. [2]Cancer Early Detection Advanced Research Institute, Oregon Health & Science University, Portland, OR, USA. [3]Scale Biosciences, San Diego, CA, USA. [4]Vollum Institute for Neuroscience, Oregon Health & Science University, Portland, OR, USA. [5]Knight Cancer Institute, Oregon Health & Science University, Portland, OR, USA. [6]Knight Cardiovascular Institute, Oregon Health & Science University, Portland, OR, USA. ✉e-mail: adey@ohsu.edu

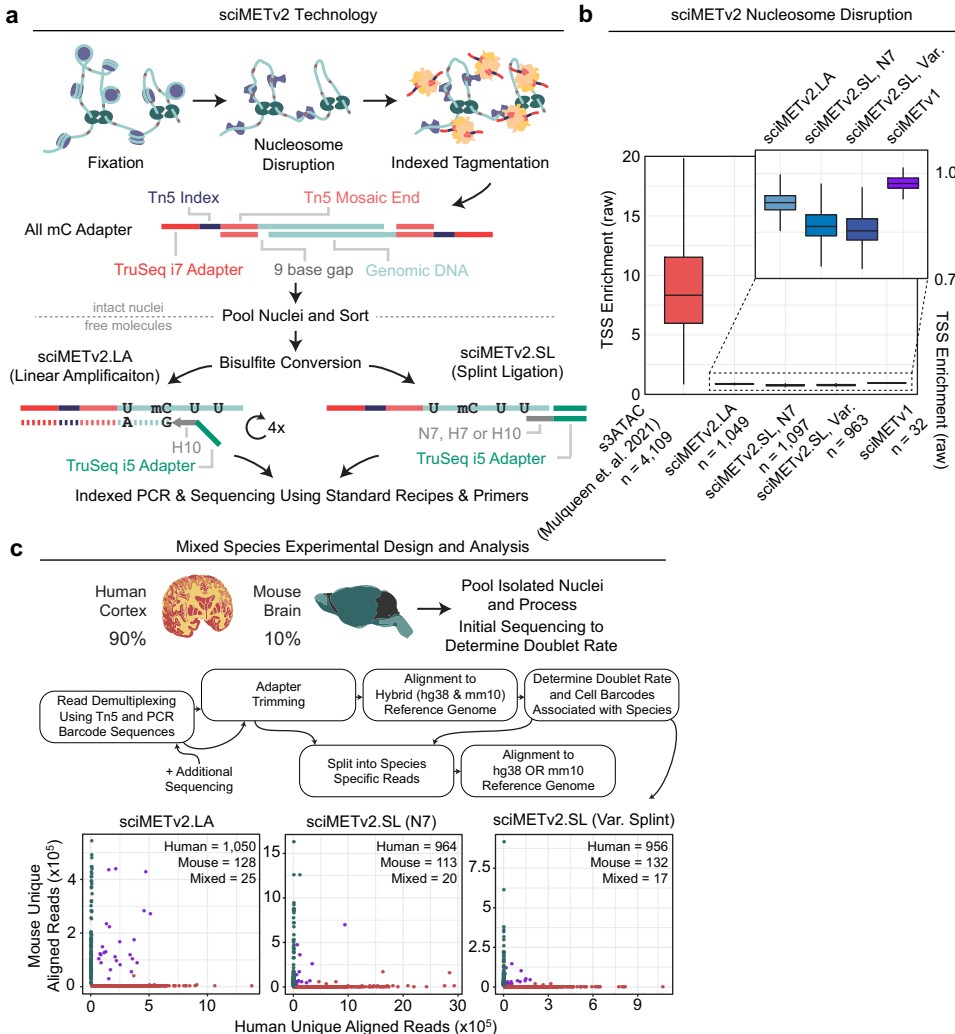

**Fig. 1 | sciMETv2 method and experimental design. a** Molecular workflow for the sciMETv2 technologies. **b** Nucleosome disruption effectiveness as measured by raw transcription start site (TSS) enrichment. Tagmentation of intact nuclei preserves TSS enrichment which is ablated by nucleosome disruption in sciMET assays. For all boxplots, boxes indicate median, 25th and 75th percentiles with min and max lines as 1.5x interquartile range. **c** Experimental design for the human and mouse mixing experiments, the analysis workflow, and species alignment rates for a subset of the sequencing reads in order to determine doublet rates and identify species specific cell barcodes.

genomic DNA contained within the nucleus for indexed transposase-based library preparation (tagmentation) using C-depleted adapters, while maintaining nuclear integrity. Nuclei are then pooled and sorted with a limited number of pre-indexed nuclei per well followed by bisulfite conversion, reverse adapter incorporation using random priming similar to post-bisulfite adapter tagging (PBAT)[11], indexed PCR amplification, and sequencing. This technique enabled ~1000 single-cell methylomes to be produced from a single 96-well plate-based experiment, without requiring any cells to be processed in an individual reaction compartment, cutting costs and enabling high throughput.

However, the original sciMET technology suffers from lower per-cell coverage when compared to other techniques that rely on one or more processing steps to be carried out in individual wells[6]. Furthermore, sciMET sequencing libraries require a custom sequencing recipe and custom sequencing primers, making the technology inaccessible to users that are not able to implement these components.

Here, we directly address each of these shortcomings by developing two complementary technologies: sciMETv2.LA (linear amplification), a highly-optimized version of the original workflow, and sciMETv2.SL (splint ligation), a rapid workflow that involves far less

processing time, and a reduction in reagent usage resulting in a 10-fold lower cost per cell.

## Results

### SciMETv2 enables high-throughput and high-coverage single-cell methylomes from brain tissue

The first optimization shared by both sciMETv2 workflows is the use of fully-methylated indexed tagmentation adapters that are modeled after our recently-published s3 complexes (symmetrical-strand single-cell combinatorial indexing; Supplementary Data 1)[12]. These include the standard Read 2 sequencing primer, a unique index, and the mosaic end (ME) sequence recognized by the Tn5 transposase to form tagmentation complexes (Fig. 1a), enabling the use of standard sequencing recipes. The second major improvement shared between the techniques is the use of optimized nucleosome disruption methods which we previously described for s3-WGS (for single-cell Whole Genome Sequencing) and s3-GCC (single-cell Genome sequencing plus Chromosome Conformation)[12]. These improvements include a reduction in formaldehyde concentration, thus reducing the amount of fixation which may impede tagmentation efficiency while achieving enough fixation to preserve nuclei integrity during the detergent (SDS)

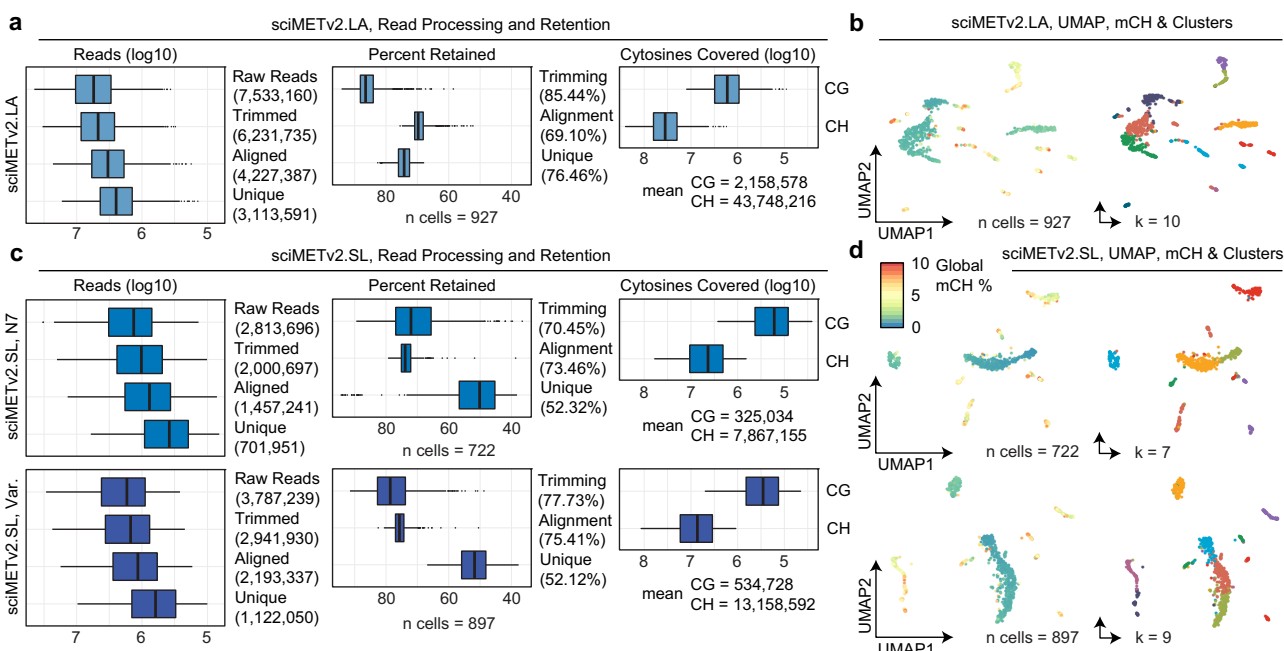

**Fig. 2 | sciMETv2 technical performance. a** Read processing and retention for sciMETv2.LA, mean values in parentheses. **b** UMAP projection colored by global CH methylation levels (left) or cluster (right). **c** Read processing and retention for sciMETv2.SL experiments. **d** UMAP projection colored by global CH methylation levels (left) or cluster (right). For all boxplots, boxes indicate median, 25th and 75th percentiles with min and max lines as 1.5x interquartile range.

based nucleosome disruption, which was also reduced. Together these optimizations achieve greater tagmentation efficiency, which translates to increased per-cell coverage, with a comparable coverage uniformity as assessed by the ablation of transcription start site (TSS) enrichment (Fig. 1b). Finally, in addition to a series of minor optimizations, for sciMETv2.LA specifically, we leverage H-bases (33% A, 33% C, 33% T) in the random priming region, as in snmC-seq2, which provides higher specificity for bisulfite converted DNA which largely lacks any cytosines[6].

We deployed sciMETv2.LA to profile a skewed mix of nuclei prepared from whole mouse brain (~10%) and human cortex (~90%, middle frontal gyrus) in order to assess performance on primary tissue samples (Fig. 1c). The mouse cells were included to assess the cell doublet rate as well as assess any cell-cell crosstalk (Fig. 1c), resulting in 1203 cells: 1050 human, 128 mouse and 25 mixed. Notably this results in a higher doublet rate than the original sciMET workflow (19.5% vs <10%). This is due to depositing 22 sort events per well for downstream processing for which only half were actual viable cells in the original workflow; whereas we achieved a far greater sort event viability with sciMETv2, with nearly all events producing cell profiles. However, the doublet rate was very near the expected count if 100% of events were viable (22.9%), suggesting that the doublet rate is tunable and can be proportionately reduced by decreasing the deposited cells per well. The second component we assessed from the mixed-species experiment was the presence of cell-cell crosstalk which we determined to be <1% across cells for each respective species.

Additional sequencing was then performed, retaining only reads assigned to the human cell indexes producing an average of 85.44% of reads retained after adapter trimming and 69.1% of reads aligned, comparable to the reported alignment rate of snmC-seq2 at 64.7%[6]. PCR duplicate reads were then removed, retaining a mean of 76.46% of reads, for 3,113,591 mean unique reads per cell (Fig. 2a), a 14.2-fold improvement over the original sciMET technology at the same level of sequencing saturation. In total a mean of 2.2 million unique CGs were assessed, with a maximum of 12,331,231 covered CGs, or ~44% of the CG dinucleotides present in the human genome, as well as 43.7 million unique CH sites. Given the abundant methylation in the CH context in

neuronal cell types, we assessed CH methylation levels in 250 kbp windows followed by single value decomposition (SVD) and clustering; producing 10 distinct clusters visualized in a UMAP projection (Fig. 2b; Supplementary Data 2).

## A rapid version of sciMETv2 provides sufficient methylome coverage with a reduction in processing time and costs

We next addressed the final shortcoming of sciMET: multiple rounds of linear extension; a lengthy process that uses large amounts of several enzymes. We elected to leverage a modified version of the SRSLY technology, originally developed for preparing sequencing libraries from single-stranded ancient DNA or cell-free DNA specimens[13], to replace the linear extension steps for the addition of the Read 1 primer sequence. This technique uses a randomer-annealed splint oligo to ligate the desired adapter, along with the addition of single-stranded DNA binding (SSB) protein to enhance efficiency. A similar approach was also recently described for direct adapter ligation for single-cell DNA methylation profiling, scSPLAT[7], though without the use of SSB and leveraging different adapter designs. After ligation, indexed PCR is performed prior to pooling and sequencing. We evaluated this technique on the same mix of human cortex and mouse brain nuclei as used in the previous sciMETv2 experiments to produce 1097 QC-passing single-cell methylomes, 964 of which were derived from the human specimen, with 722 passing final QC thresholds. The mean unique, usable sequence reads per cell was 701,951, notably lower than that of sciMETv2.LA, after trimming (mean 70.45% retained), alignment (mean 73.46% aligned), and duplicate removal (mean 52.12% retained); providing a mean coverage of 325,035 CG sites and 7,867,155 CH sites per cell (Fig. 2c). Despite the lower coverage, we were able to identify 7 distinct clusters (Fig. 2d).

To optimize performance, we next tested three different splint designs on the same tissue mixture including the original splint, N7, and two splints comprised of H bases (33% A, 33% C, 33% T) with a length of 7 or 10 bases (H7 and H10). On aggregate, the preparation produced a comparable doublet proportion and cell number (1105 total, 956 human), with slightly improved trimming retention (mean 77.43%), alignment rate (mean 73.46%), and a comparable unique read

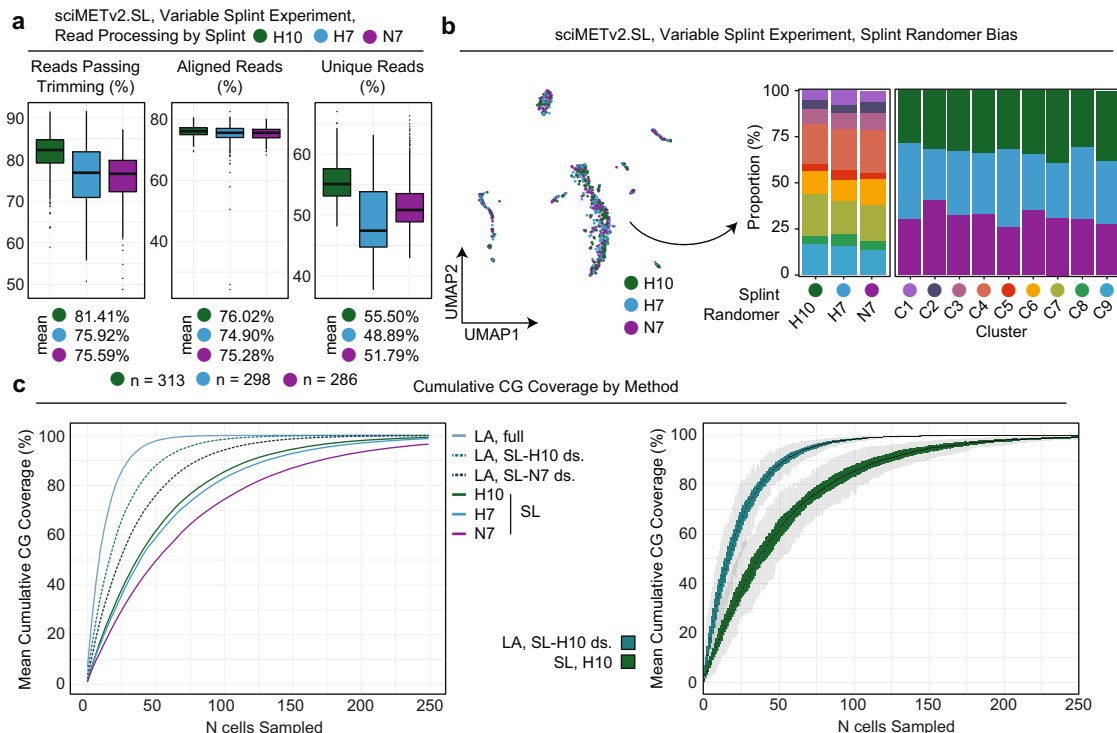

**Fig. 3 | sciMETv2 splint ligation optimization and performance. a** Read processing and retention broken down by the three splint designs in the variable splint sciMETv2.SL experiment. For all boxplots, boxes indicate median, 25th and 75th percentiles with min and max lines as 1.5x interquartile range. **b** UMAP colored by variable splint (left) and composition of clusters by splint and splint by clusters (right). **c** Cumulative coverage from sampled cells for each method over 100 iterations. Mean values for iterations (left) are shown and include two down sampled variants of the sciMETv2.LA dataset to match the raw read count of the sciMETv2.SL datasets using the H10 (LA, SL-H10 ds.) or N7 (LA, SL-N7 ds.) splints. The distribution of coverage across iterations is shown for the LA, SL-H10 ds. and the SL-H10 datasets (right).

percentage (mean 52.32%), for an average usable read count of 1,122,050 with 534,728 CG and 13,158,592 CH sites covered (Fig. 2c). While coverage is less than that of the sciMETv2.LA method, this gap is reduced when the sciMETv2.LA dataset is down sampled to the same raw reads per cell as this experiment, which produced coverage at a mean of 1,197,590 CG sites per cell. After matrix generation and filtering, 897 cell profiles were retained which produced 9 distinct clusters (Fig. 2d). We next broke down the experiment into cells produced by each splint sequence. Little difference was observed between the two 7 bp splint sequences (Fig. 3a); however, the H10 splint produced the highest read retention rate after adapter trimming (mean 81.41%), and the highest unique read percentage (mean 55.50%). We evaluated the potential impact of splint sequence on downstream processing which revealed minimal bias with respect to cluster composition across splints and splint composition across clusters (Fig. 3b).

A key component of single-cell DNA methylation analysis is the ability to derive full-coverage methylomes for distinct cell types that are identified. To compare performance across methods, we iteratively sampled a set of 250 cells from each method and calculated the CG methylome coverage accumulation as each cell's coverage was added (Fig. 3c). As expected, the sciMETv2.LA method achieved the highest methylome coverage with the fewest cells, requiring an average of 21 cells to achieve 80% CG-methylome coverage across iterations. However, each cell from the sciMETv2.LA preparation had far greater raw reads sequenced per cell (Fig. 2a, b). We therefore randomly down sampled the raw read counts to be comparable to the sciMETv2.SL preparations using the N7 (worst-performing) and H10 (best-performing) splints. The LA down-sampled data still produced the greatest coverage per-cell, achieving 80% coverage at an average of 38 cells versus an average of 84 cells for the SL prep using the H10 splint, equating to a 2.2-fold improved efficiency using the LA

method over SL at a comparable read count. This difference is considerable; however, the preparation time and costs are greater per-cell for the LA method at 10 h versus just over 2, and ~$2300 per plate versus ~$230. This equates to a 10-fold cost reduction per cell in preparation costs meaning smaller experiments may achieve a comparable cost point depending on sequencing costs (Supplementary Fig. 1; Note: the LA preparation is more expensive than the original sciMETv1 workflow due to increased costs for Klenow fragment).

## sciMETv2 provides cell subtype resolution in the human brain with no discernible method bias

We next combined data across experiments by directly merging 250 kbp window CH methylation matrices retaining 2546 cells after filtering (Methods), followed by SVD, clustering, and UMAP visualization (Fig. 4a). Global CH methylation levels revealed a distinct split between clusters, representing neurons and non-neuron cell types (Fig. 4b). We next calculated CG methylation levels at the promoter regions (−1500 to +1000 bp of TSS) of all genes by cluster (Supplementary Data 3), and then aggregated these data into cell type categories based on marker gene methylation (Fig. 4c). This resulted in three oligodendrocyte clusters, a single cluster each for astrocytes and microglia, six excitatory neuron clusters, and three inhibitory neuron clusters. We then assessed neuronal cell types by including gene body CH methylation levels for subtype markers[2] (Supplementary Data 4, 5), revealing distinct methylation patterns delineating excitatory and inhibitory neuron subtypes (Fig. 4c). Cell-cell methylation variability was then examined both within and across cell types. We performed an all-by-all pairwise calculation of CG methylation status concordance across all shared CG sites within promoter regions (Fig. 5a). Concordance was generally high within cell type clusters with a mean similarity >90% for all clusters other than one of the excitatory neuron

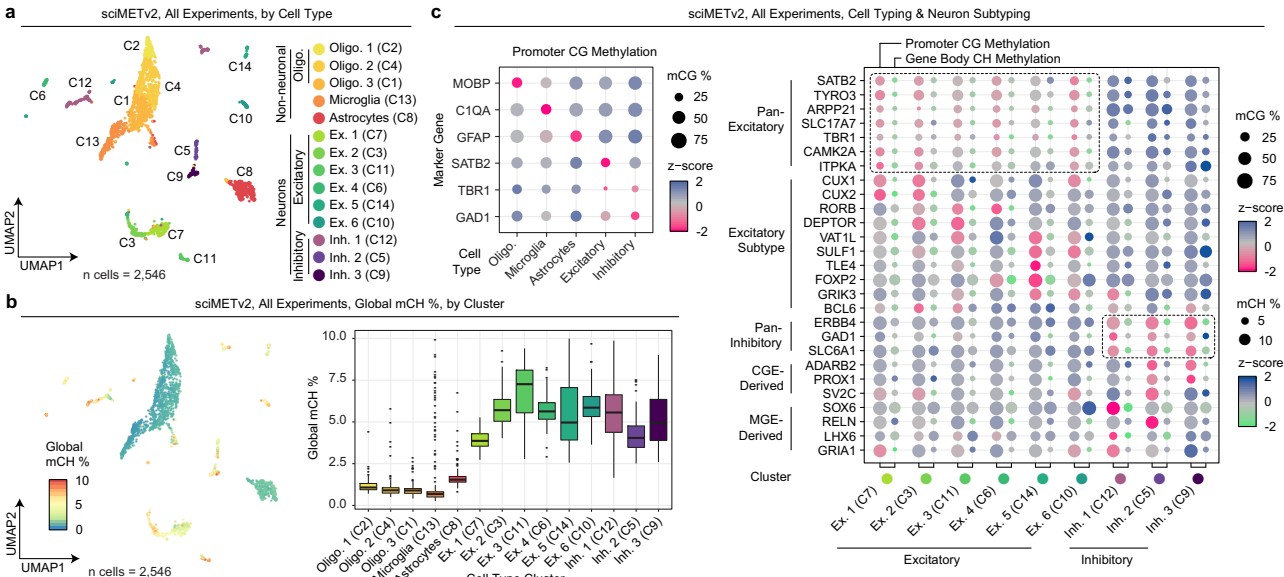

**Fig. 4 | sciMETv2 cell type analysis and method bias assessment. a** UMAP projection of the three sciMETv2 preparations integrated together colored by cluster. **b** Global CH methylation levels per cell in the UMAP projection (left) and by cluster (right). For all boxplots, boxes indicate median, 25th and 75th percentiles with min and max lines as 1.5x interquartile range. **c** Promoter CG methylation levels of marker genes for cell-type-aggregated clusters (left) and both promoter CG methylation and gene body CH methylation for neuronal marker genes and subtype markers. Dashed lines represent clusters showing evidence of positive gene regulation for excitatory (top) or inhibitory (bottom) marker genes.

clusters (Ex. 5). Notably, this cluster was visually split between two locations on the UMAP projection (Fig. 4a), suggesting a higher clustering granularity may be warranted. When assessing variability across clusters, the mean similarity was lower across all comparisons with the greatest difference residing between excitatory neurons and non-neurons.

To examine the utility of cell type methylomes to assess additional epigenetic features, we leveraged the aggregate methylomes of each cell type to examine cell-type-specific methylation patterns at DNA binding motifs for factors known to be methylation-sensitive (Fig. 5b). We first assessed the CTCF motif, which did not exhibit cell type specificity, as all cell types show a marked decrease in methylation centered on the motif. Given the role of CTCF in genomic topological domain establishment and stability, the consistency across cell types was expected[14]. In contrast, the RE1 motif sequence, which corresponds to the REST protein, demonstrated a marked decrease in CG methylation specific to glial cell populations. This is consistent with the role of REST as a transcriptional repressor often found at genes expressed in neuronal lineages[15].

Finally, experimental bias was then evaluated on the merged dataset without any normalization, revealing slight skewing in the UMAP visualization. However, when assessing the cell type composition for each preparation, there was very little variance, with each method producing similar proportions for each cluster. Similarly, each cell type cluster was represented by similar proportions of cells from each experiment (Fig. 2d). We next performed an additional assessment of bias by directly integrating our data with cells produced using snmC-seq2 from two frontal cortex specimens at ages 25 and 58 with 76 and 75 cell profiles, respectively[6]. SnmC-seq2 reads were downloaded and processed to produce the same CH methylation matrixes as our sciMETv2 datasets and integrated as detailed above, again with no normalization. Notably, the snmC-seq2 cells were sorted for NeuN+, enriching heavily for neurons. We therefore only assed cluster composition for neuronal cell types. This produced a comparable breakdown across cell types for each method (Fig. 2e), with the exception of one single cluster that was comprised primarily of cluster C7 on the sciMETv2 integrated analysis and identified as an excitatory neuron

subtype. This cluster may be present only in the sciMETv2 preparations due to region specificity (sciMETv2: middle frontal gyrus vs. snmC-seq2: prefrontal cortex). Finally, we performed an additional experiment where we leveraged three 96-well plates of uniquely barcoded methylated Tn5 complexes on a middle frontal gyrus specimen from a separate individual, using the sciMETv2.LA workflow. For this preparation we deposited 10 events per well per tagmentation plate (30 total) across a single 96-well PCR plate which produced 2071 cell profiles and an estimated doublet rate of <7.5%. Again, without performing any batch correction, cells were assigned across all clusters regardless of the preparation, though with a much larger proportion of glial cells from the new specimen, likely due to sampling biases (Fig. 6c).

**Distinct cell types can be identified using CG methylation alone**

DNA methylation is most commonly associated with CG sites and their unique regulatory role in mammalian genomes, whereas methylation of cytosines in the CH context is generally rare, restricted to embryonic stem cell states and neurons[16]. To demonstrate the capabilities of sciMETv2 to assess other tissue types that lack CH methylation, we reanalyzed the data from our sciMETv2 experiments using exclusively CG sites employing non-overlapping 50 kbp windows (Fig. 7a). A smaller window size was chosen due to the smaller typical feature size of CG elements (e.g., CG islands) versus the larger features in the CH methylation context (i.e., gene bodies). For the sciMETv2.LA preparation that produced higher coverage, 1012 cells passed filters (Methods), which were then taken through SVD, clustering, and UMAP visualization, producing 7 distinct clusters. Similarly, the two sciMETv2.SL preparations produced 4 and 5 distinct clusters for the N7 and variable splint preparations, respectively; however, these preparations had fewer cells passing coverage filters (329 for N7 splint and 558 for varied splint). We next aggregated the three datasets, totaling 1899 passing cells which produced 10 distinct clusters that corresponded to the clusters identified by CH methods, including three Oligodendrocyte clusters, a single cluster each for microglia and astrocytes, and two clusters for both excitatory and inhibitory neurons (Fig. 7b). As expected, the reduction in clusters was primarily in the

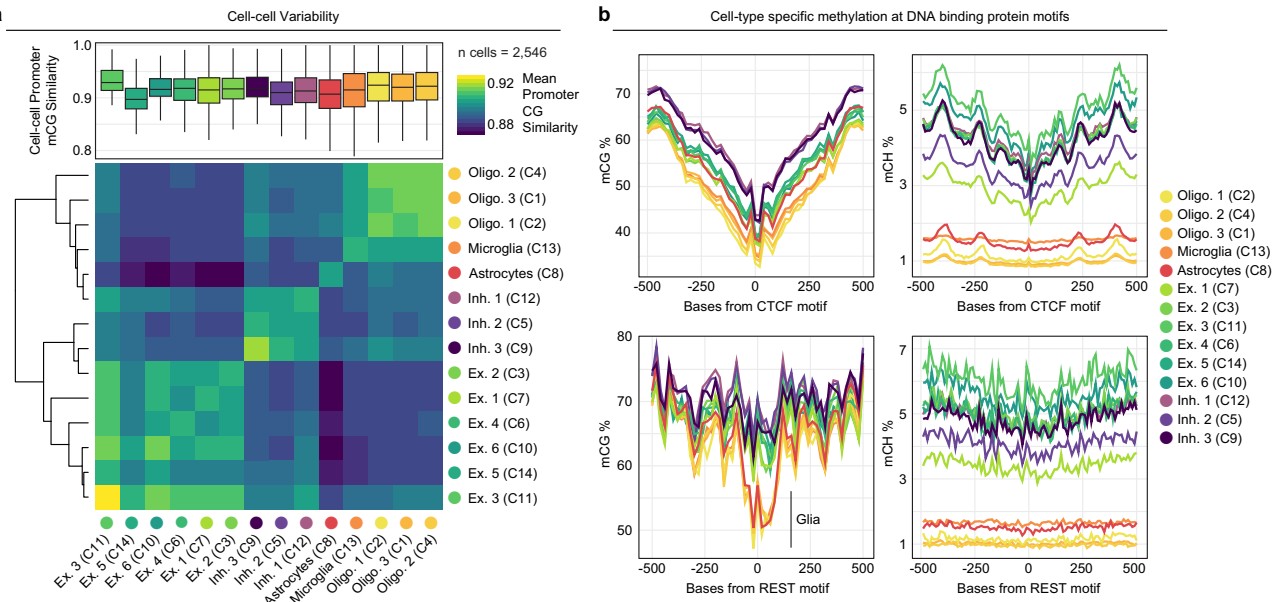

**Fig. 5 | Cell-cell variability and motif analysis of sciMETv2 data. a** Cell-cell variability as measured by all-by-all methylation similarity at promoter CG sites within each cluster (top), and the mean similarity for cells of different cell types represented as a heatmap (bottom). For all boxplots, boxes indicate median, 25th and 75th percentiles with min and max lines as 1.5x interquartile range. **b** Assessment of cell-type-specific methylation patterns for CG (left) and CH (right) centered on motifs for CTCF (top) and REST (RE1, bottom). Reduced CG methylation at RE1 sites in glial cell populations is noted.

neuronal populations where CH methylation is abundant and drives cell subtyping. Taken together, sciMETv2 provides sufficient power to assess cell types within tissues using methylation in the CG context alone.

## Discussion

In summation, we present two technologies to produce high-throughput single-cell DNA methylation libraries: a high-coverage, robust workflow capable of >14-fold improved coverage when compared to the preceding technology, and a rapid version with lower coverage yet further reduced costs and time. We demonstrate that both techniques provide sufficient coverage to assess complex tissues using either CH (neurons) or CG methylation, which is relevant for most other tissue types. Both strategies also leverage standard sequencing workflows and include multiple safe stopping points. The primary remaining challenge of sciMETv2.LA is that the read insert size is low, with many read pairs fully overlapping. This is likely due to damage during bisulfite conversion, making the exploration of enzymatic conversion methods an appealing alternative to potentially achieve a larger fragment size and thus more unique coverage per aligned read. The sciMETv2.SL variant produces less coverage per-cell than the sciMETv2.LA variant at comparable sequencing coverage; however, it is substantially cheaper and faster, making it more favorable as sequencing costs decrease. Furthermore, the simplicity of the splint ligation workflow makes it more appealing for large numbers of cells, motivating further optimization to achieve a similar coverage per cell compared to the linear amplification variant.

## Methods

### Ethical statement

Human tissue used in this study was obtained as de-identified specimens from the Oregon Brain Bank which is overseen by the OHSU Institutional Review Board and consented for genetic data sharing and genomic data under restricted access for research purposes only. Raw data from specimens are deposited in the Neuroscience Multi-omic Data Archive (NeMO) under restricted access with processed data that does not contain genomic variants openly accessible. Mouse specimens were obtained from discarded tissue and did not require ethical approval.

### Statistics and reproducibility

No statistical method was used to predetermine sample size. Cell deposition for second-tier indexing was determined based on empirical data and that produced a <5% collision rate for the sciMETv1 assay. All cells meeting a minimal coverage threshold and below 10% methylated CH bases were included in this study. The experiments were not randomized. The Investigators were not blinded to allocation during experiments and outcome assessment.

### sciMETv2 nucleosome disruption and tagmentation

A detailed step-by-step protocol is provides as Supplementary Note 1 in the Supplementary Information associated with this manuscript.

Human middle frontal gyrus samples were obtained from the Oregon Brain Bank where they were stored by immediately placing dissected tissue cassettes in a −80 °C freezer. Mouse (C57BL6, female, P56) specimens were obtained as discarded tissue and preserved by flash-freezing in a liquid nitrogen bath. A portion of tissue was isolated using a sterile razor blade and placed in a dish containing ice cold Nuclei Isolation Buffer (NIB, as in Thornton et. al. 2021; available as ScaleBio Part. No. 230041)[17]. The slurry was then homogenized with a Dounce homogenizer using a loose pestle for 7 strokes, incubating 10 min on ice, and repeating the homogenization with a tight pestle for 7 strokes. Nuclei were spun down by centrifugation at 500 xG for 5 min at 4 °C, the supernatant removed, and then resuspended in 200 μl ice cold NIB. Nuclei were then carried through nucleosome disruption according to Mulqueen et. al. 2021 using buffers available from ScaleBio[12]. Nuclei were first fixed at room temperature in 1 million nuclei aliquots using 1 mL Fixation Reaction Mastermix (FRM, ScaleBio Part No. 230011) for each aliquot. Nuclei were incubated at room temperature for 10 min with occasional mixing. Fixation was stopped by adding an equal volume of Stop Master Mix (STP, ScaleBio Part No. 230021) and incubated on ice for 5 min. Nuclei were spun down at 500xG at 4 °C for 5 min and resuspended in Nucleosome Depletion

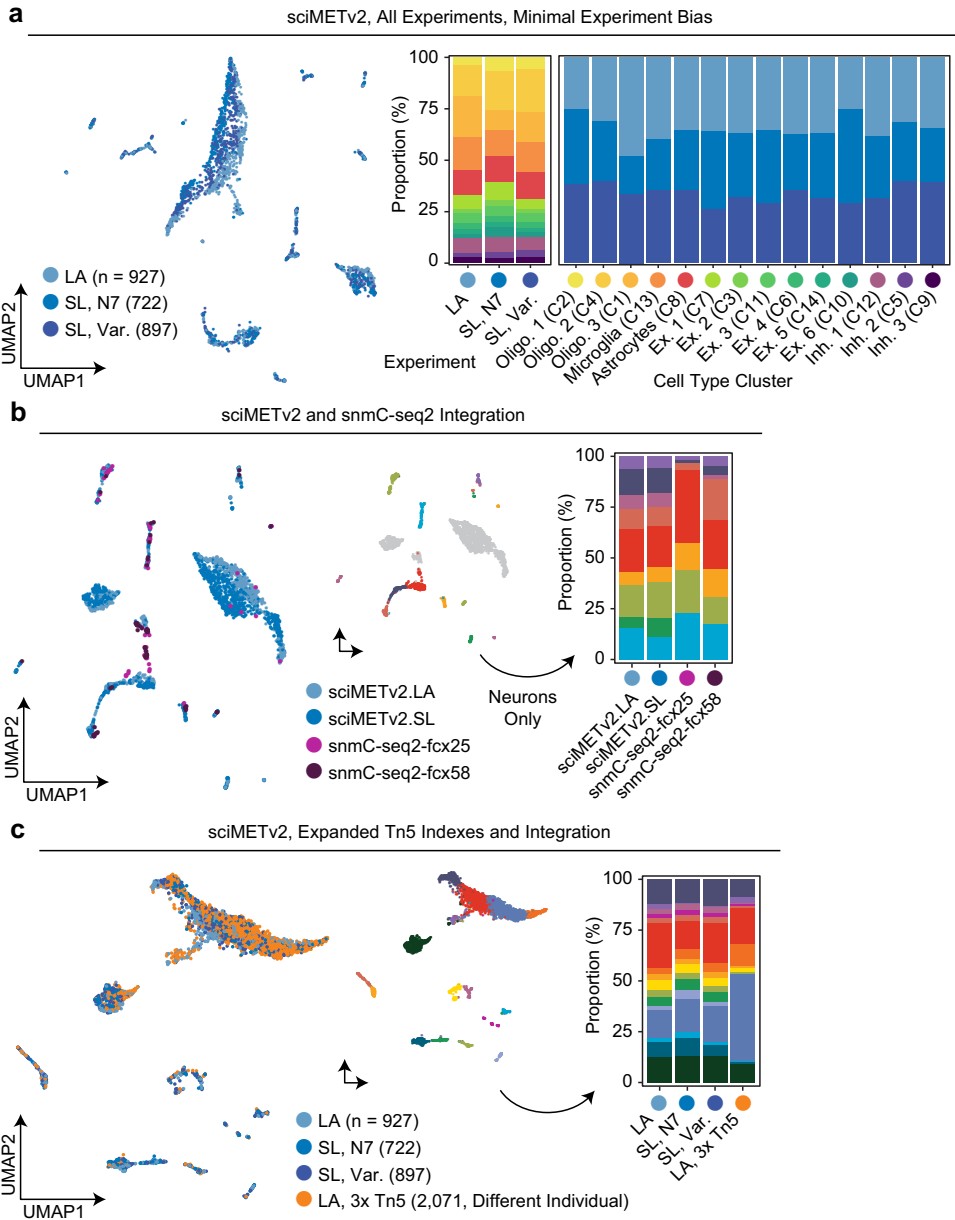

**Fig. 6 | Assessment of bias between experiments and technologies. a** Integrated UMAP colored by sciMETv2 experiment (left) and the cluster breakdown for each experiment as well as experiment breakdown for each cluster (right). **b** Integration with human frontal cortex snmC-seq2 data projected into a UMAP and colored by assay (left), and by clusters (middle, non-neuron clusters are grayed out), with the cluster composition of each method (right). **c** Integration with a sciMETv2.LA dataset produced using three sets of 96 indexes at the tagmentation step performed on a different individual. Cells from each method were present in each cluster, though greater counts of glial cells were observed in the new preparation.

Mastermix (NDM, ScaleBio Part No. 230031) and incubated at 37 °C for 20 min. Reactions were then spun down at 500xG at 4 °C for 5 min and resuspended in 200 μL NIB. Aliquots were combined at this point.

Tagmentation was carried out in a 96-well plate. Each well contained 5000 nuclei, 2.5 uL 4X-Taps-TD (available from ScaleBio prior to commercial availability; 33 mM TAPS pH = 8.5 [Sigma, Cat. T5130], 66 mM KOAc [Sigma, Cat. P1190], 10 mM MgOAc [Sigma, Cat. M5661], 16% DMF [Sigma, Cat. D4551]), 2 uL 5 μM Tn5 complexes with methylated Cs using the oligo sequences detailed in Supplementary Data 1 (available from ScaleBio upon request prior to commercial availability) and enough NIB to fill the volume to 10 uL. Note the 4X-Taps-TD is to be made fresh, the day of the experiment. The plate was then incubated for 15 min at 55 °C. Afterwards it was put on ice. For the preparation leveraging 288 tagmentation indexes, three 96-well plates were prepared

identically but with the use of three different sets of 96 barcoded Tn5 complexes.

All wells were pooled and they were run through a 40 μm cell strainer. 3 uL 5 mg/mL DAPI was added to stain the nuclei. They were FACS/FANS sorted at 15–30 nuclei per well into 96-well plates with each well containing 1 uL M-Digestion Buffer (Zymo Research: Cat. D5021-9), 0.07 μL reconstituted Qiagen Proteinase K (Qiagen: Cat. 19131), and 0.93 μL dH2O. These post-sort plates were then spun down at 500xG at 4 °C for 10 s. They were then incubated at 50 °C for 20 min.

## sciMETv2 bisulfite conversion

Bisulfite conversion was adapted from the snmC-seq2 protocol[6], 1 bottle of CT Conversion Reagent was reconstituted with 7.9 mL M-Solubilization Buffer and 3 mL M-Dilution Buffer. The bottle was shaken vigorously to dissolve. After the reagent was dissolved, 1.6

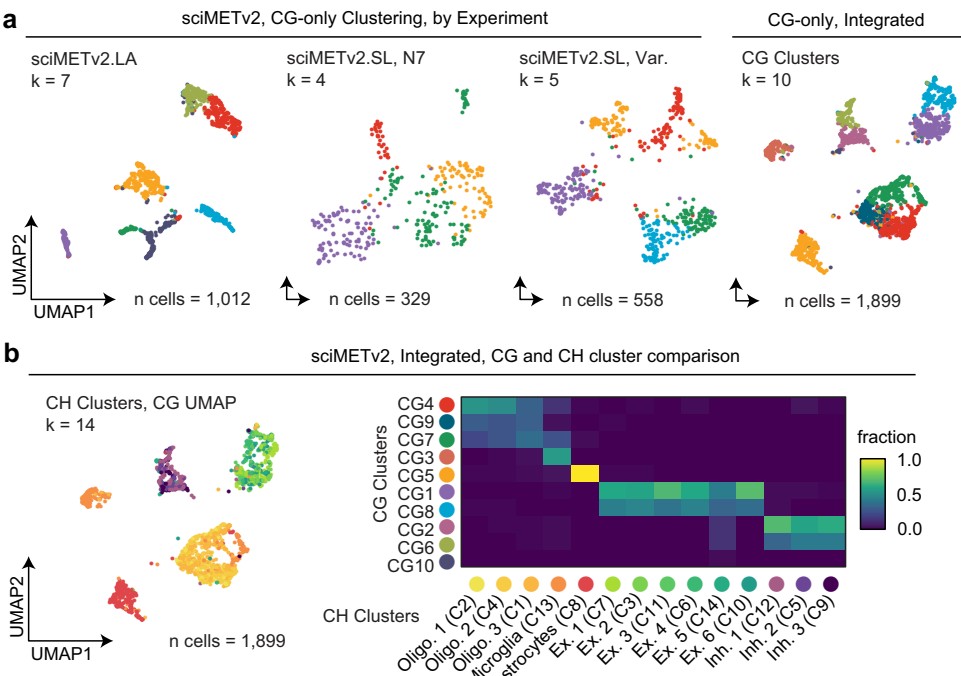

**Fig. 7 | Analysis of sciMETv2 datasets using CG methylation alone. a** UMAP projections colored by cluster for each sciMETv2 experiment when analyzed using CG methylation alone and for the integrated sciMETv2 dataset processed (right). **b** CG based UMAP on the integrated dataset as in **a**, with cells colored by CH methylation clusters that were previously determined and a confusion matrix comparing the CG (*y*-axis) and CH (*x*-axis) cluster identities.

mL M-Reaction Buffer was added. After a quick spin down of the post-sort plates, 15 μL of the reconstituted CT Conversion Reagent was added to each well. The plates were then incubated at 98 °C for 8 min, then at 64 °C for 3.5 h with a final 4 °C hold overnight.

To clean the bisulfite reactions 80 μL M-Binding Buffer was added to each well and the entire volume of each well was then transferred to a 96-well Zymo-Spin I-96 Plate (Shallow well). The column plates were then spun down at 2200 xG for 8 min. Flow-through was discarded and 100 μL M-Wash Buffer was added to each well and spun at 2200 xG for 8 min or until all Wash Buffer came out. Flow-through was discarded and 50 μL M-Desulphonation Buffer was added to each well. The plates were incubated for 15 min at room temperature. Afterwards, they were spun at 2200 xG for 8 min, discarding the flow-through. 200 μL M-Wash Buffer was added to each well and spun out at the above speed and time. After discarding the flow-through, the plates were spun again to completely dry the columns. Bisulfite converted DNA was eluted by adding Buffer EB that was preheated at 55 °C.

### sciMETv2.LA
Bisulfite converted DNA was eluted by adding 25 μL Buffer EB to each column. The plate was incubated for 4 min at 55 °C by placing on top of a 96-well plate containing the following in each well: 17.8 μL dH2O, 5 μL 10X NEB Buffer 2.1, 2 μL 10 mM dNTPs and 0.2 μL 100 μM 9H Random Primer. The assembled plates were spun at the above speed and time to elute the DNA. This is the linear amp (LA) plate.

The LA plate was then heat shocked at 95 °C for 45 s and then placed on ice until cool. 10 units of Klenow exo⁻ was added and placed on a thermocycler with the following incubations: 4 °C for 5 min, ramp of 1 °C every 15 s until 37 °C is reached and incubate for 90 min at 37 °C, then hold at 4 °C. This is the first LA cycle.

The plate was then heat shocked again at 95 °C for 45 s and placed on ice. The following was then added to each well of the plate: 0.1 μL 100 μM 9H Random Primer, 1 μL 10 mM dNTPs, 0.5 μL 10X NEB Buffer 2.1, 1.65 μL dH2O and 2 μL of Klenow exo- (5U/uL). The above thermocycling protocol was run again. This is the second LA cycle. This cycle was repeated twice more for a total of 4 LA cycles.

After a 1.1:1 (SPRI to template ratio) SPRI cleanup, the plate was eluted with 21 μL Buffer EB into a plate containing the following (final volume is 50 μL): 25 μL Q5U 2X Master Mix, 2 μL 10 μM TruSeq i5 primers, 2 μL TruSeq i7 primers and 0.5 μL EvaGreen 100X. The plate was placed on a qPCR machine using the following program: 95 °C 2 min initial activation, 94 °C 80 s denaturation, 65 °C 30 s annealing, 72 °C 30 s extension, 72 C 10 s plate read. Once most wells began to plateau, the plate was pulled. All wells were pooled for cleanup and library QC.

### sciMETv2.SL
Bisulfite converted DNA was eluted by adding 5 μL preheated Buffer EB to each column in the plate. The assembled plates were incubated for 4 min at 55 °C by placing on top of a 96-well plate containing 1 μL 20 ng/uL ET-SSB.

The plate was then heat shocked at 95 °C for 3 min and placed on ice for 2 min. 1 μL 0.75 μM pre-annealed P5 adapter was then added to each well. The ligation master mix (4 μL 50% PEG 8000, 0.75 μL SCR Buffer, 0.1 μL 1 M DTT, 0.1 μL 100 mM ATP, 0.125 μL T4 PNK [10,000 U/mL, NEB], and 0.125 μL T4 Ligase [2,000,000 U/mL, NEB]) was added to the plate at room temperature. Note: The splint ligation reaction relies on the ligation master mix being set up at room temperature and was carried out according to Kapp et al. 2021 protocols[13]. No cleanup was performed between ligation and PCR.

For indexing PCR, the following was added to each well of the ligation plate: 10 μL 5X VeraSeq GC Buffer, 2 μL 10 μM dNTPs, 1.5 μL VeraSeq Ultra Enzyme, 24 μL dH2O, 0.5 μL 100X EvaGreen, 1 μL 10 μM TruSeq i5 primers, 1 μL 10 μM TruSeq i7 primers. The following cycling program was used: 98 °C 30 s initial activation, 98 °C 10 s denaturation, 57 °C 20 s annealing, 72 °C 30 s extension, 72 °C 10 s plate read. Once most wells began to plateau, the plate was pulled. All wells were pooled for cleanup and library QC.

### Computational analysis
All code used to process and analyze data can be found here: github.com/adeylab/sciMETv2, except for the initial demultiplexing

which can be found here: github.com/adeylab/unidex, or other wrapper functions here: github.com/adeylab/scitools.

### Read processing and alignment

Raw sequence reads were demultiplexed using unidex (github.com/adeylab/unidex) with a custom mode that uses the three indexes: forward PCR, reverse PCR, and tagmentation index, allowing for a hamming distance of 2. The first 10 bases of read 1 (random priming region), bases 1-8 (index) and 9-29 of read 2 (mosaic end) were all trimmed during this processing. Reads were then carried through sciMET_trim.pl, a part of the sciMET GitHub repository (github.com/adeylab/sciMETv2), using '-e 10' which uses trimmomatic[18] to trim adapter sequences in single-end mode, followed by removing the last 10 bases of read 2 since they may include the random primer region. Reads were then aligned using bismark (v0.23.1)[19] with the wrapper script: sciMET_align.pl which uses the '–pbat' option for read 1 and the default directional option for read 2. Initial alignment was to a reference genome containing both the human and mouse genome to identify species identity and assess doublet rates and the presence of any cross-cell contamination using scitools barnyard-compare (github.com/adeylab/scitools)[20] on the bam file after barcode-aware duplicate read removal using sciMET_rmdup.pl. Once species were identified, trimmed reads for those cells were then split to the respective species-specific read files using sciMET_speciesSplit.pl and then aligned to their respective genomes following the same workflow.

For the preparation that leveraged 288 tagmentation indexes (3 × 96-well plates) we optimized the processing and alignment pipeline with updated scripts, also available in the sciMET GitHub repository. The new trimming workflow uses TrimGalore (v0.6.5), which leverages CutAdapt (v4.1)[21], within the wrapper script sciMET_trim.pl, which performs a pair-wise read trim. Properly paired, trimmed reads were then aligned using bismark[19] with the wrapper script sciMET_align_pe.pl which swaps reads 1 and 2 and then performs a paired alignment using the '–local' option. We found that these variants provided a notable improvement in processing time and achieving a comparable alignment rate.

TSS Enrichment was calculated using averaged 200 bp windows 1000 bp away from TSSs as background and the 200 bp centered on the TSS as signal. Shotgun WGS data produces a value near 1 (0.944) using tagmentation-based shotgun sequencing of the human GM12878 cell line versus other TSS Enrichment methods that often produce values much higher. This method was chosen as tagmentation of intact nuclei without nucleosome disruption (*i.e.* ATAC-seq) produces high TSS Enrichment and the goal of nucleosome disruption is to ablate this signal and achieve a value at or below 1.

### Methylation calling and analysis

Species-specific aligned and duplicate removed bam files were processed using the wrapper: sciMET_extract.pl which uses the bismark methylation extraction tool with the following options: '–comprehensive–merge_non_CpG–single-end–no_header'. The extracted files were then split into individual chromosome files to aid in processing time and memory for future operations with a separate folder for CG and CH methylation calls. The CH methylation calls were then used to construct a mCH matrix of cells × 250 kbp genomic window using sciMET_meth2mtx.pl. The resulting matrix was filtered to first only include cells with a minimum of 75% of windows with 20 or more called CH sites followed by filtering windows to only retain those with at least 75% of passing cells with 20 or more called CH sites using sciMET_filtMtx.pl. The matrix was then taken through SVD, computing 50 components using the irlba function in R within the scitools irlba function. This matrix was then used for Louvain-based clustering with scitools matrix-pg, a wrapper for the phenograph tool[22], as well as used for visualization using UMAP[23].

Coverage per-cell for each method was assessed by performing 100 iterations of random cell selection where for 250 cells the cumulative methylome coverage was assessed. This was performed for each method and a downsampled version of the linear amplification method assessed at a comparable number of raw sequence reads to the splint ligation libraries.

Promoter methylation for genes by clusters was determined using the sciMET script: getGeneMeth.pl with the cluster identities previously determined for the full set of genes in RefGene taking the window 5 kbp upstream and 2.5 kbp downstream of the TSS. Gene body CH methylation levels were calculated using the same script but assessing the entire gene body along with 2 kbp upstream and downstream. These values were aggregated over annotated clusters and then z-scored across clusters to produce plots in Fig. 2c.

Cell-cell similarity scores were calculated based on the strategy detailed in Hui et. al. (2018)[5] where overlapping CG sites between cells restricted to regulatory regions (promoters encompassing +1.5 and −1 kbp from TSS) were assessed for shared methylation status using the sciMET_pairwise.pl script. Mean cross-cluster cell similarities were hierarchically clustered using R heatmap.2.

Motif methylation was assessed by identifying all genomic loci within the GRCh38 reference genome using Homer (v3.0)[24]. Windows were then generated in 10 bp increments out to 500 bp from the center of the motif using the sciMET_featuresToScanBed.pl script, and the mean methylation value calculated for each cluster using either CG or CH methylation calls using the sciMET_getWindowMeth.pl script.

snmC-seq2 data was downloaded from the NCBI Gene Expression Omnibus using accession "GSE112471". Reads were processed to remove adapters and short fragments using cutadapt[21] as detailed in the snmC-seq2 workflow[6] and then aligned using our processing workflow detailed above, swapping reads 1 and 2 to account for the reverse adapter configuration of snmC-seq2 vs sciMETv2. PCR duplicates were removed and methylation calls carried out using the same workflow as detailed above. The resulting CH methylation matrix was then merged with the sciMETv2 integrated matrix prior to filtering and processing.

CG-centered analyses were carried out as in the CH analysis but using 50 kbp non-overlapping windows and filtering the resulting matrix to retain cells with at least 25% of windows with 5 or more CG sites covered and windows with at least 50% of cells passing the same coverage threshold.

### Reporting summary

Further information on research design is available in the Nature Portfolio Reporting Summary linked to this article.

## Data availability

All raw sequence data has been deposited in the sciMETv2 collection of the Neuroscience Multi-omic Data Archive (NeMO), within the "BICCN_RF1_Adey [http://data.nemoarchive.org/biccn/grant/rf1_adey/adey/epigenome/sncell/sciMETv2/]" grant accession under restricted use access (human genetic data for research use only). Access can be obtained by requesting access through the NeMO portal. Methylation calls are also available from NeMO with open access under the same accession "BICCN_RF1_Adey [http://data.nemoarchive.org/biccn/grant/rf1_adey/adey/epigenome/sncell/sciMETv2/]". Raw sequence data are also accessible through the NCBI Database of Genotypes and Phenotypes under accession "phs003091.v1.p1". The snmC-seq2 data used in this work was downloaded from the NCBI Gene Expression Omnibus using accession "GSE112471". Source data are provided with this paper.

## Code availability

All code used in the analysis of and display of data are available here: https://github.com/adeylab/sciMETv2.

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

## Acknowledgements

We would like to thank other members of the Adey Lab for helpful suggestions and feedback. This work was funded by R35GM124704 (NIH/NIGMS) and RF1MH128842 (NIH/NIMH, BRAIN Initiative Cell Census Network, BICCN) to A.C.A.

## Author contributions

R.V.N. and A.C.A. devised the improved sciMETv2 techniques and wrote the manuscript. R.V.N. developed the sciMETv2 technologies and performed all experiments except the 3x Tn5 experiment performed by S.A.B.L.O assisted in sciMETv2.SL experiments. A.C.A. performed the computational analysis. R.M.M. contributed to the analysis. J.T., D.P. and F.J.S. performed Tn5 complex optimization and provided experimental advice. A.R.W. optimized the alignment pipeline. G.M. advised the REST binding analysis. A.C.A. supervised all aspects of the study. All authors reviewed and edited the manuscript.

## Competing interests

J.T., D.P., and F.J.S. are employees of Scale Biosciences. R.M.M., F.J.S., and A.C.A. are authors on licensed patents that cover the nucleosome disruption and indexed tagmentation design components of the technologies described in this manuscript (WO2018018008A1 and WO2018226708A1, granted). This potential conflict of interest for A.C.A. and R.M.M. has been reviewed and managed by OHSU. The remaining authors declare no competing interests.
