## [Peer Review File · Nature Communications]

REVIEWER COMMENTS

Reviewer #1 (Remarks to the Author):

In this manuscript the authors reported sciMETv2, a single-cell methylome sequencing method that has significant improvements in terms of the technical performance over the previous version (sciMET) developed by the same research team. With sciMETv2, the authors demonstrated a 15x improvement of genome coverage at the single cell level, by increasing the efficiency of chromatin disruption (prior to tagmentation) and the efficiency of constructing sequencing libraries reusing some technical innovations reported in their recently published S3 protocol. To simplify the experimental workflow, they also presented the data from two versions of a shorter ligation-based protocol, both look promising. By applying these protocols to nuclei from human and mouse brains, they benchmarked the technical performance, and compared the results with the published SnmC-seq2 data, which are considered the highest quality data in the field currently. Overall, this study brings the community closer to being able to perform large-scale single-cell methylome routinely. The sciMETv2 method itself, as well as several technical innovations behind this method, would be of high interests to the community of single-cell genomics. I think this manuscript is appropriate for Nature Communications.

There are a few areas whether this manuscript can be further improved, mostly from the perspective of potential users.

(1) It would be really helpful to provide an overview of the experimental workflow as a supplementary figure, including the amount of time/effort for each step in each work day, so that the potential users can decide whether this is a method feasible to implement. With that figure, the readers might be able to appreciate better the advantage of sciMETv2.SL over sciMETv2.LA, since based on the performance metrics alone, it's hard to justify sciMETv2.SL.

(2) While the authors have demonstrated cell type clustering using both mCH and mCG signatures, and also integrated sciMETv2 data with SnmC-seq2 data, grouping single cells into discrete cell-type specific clusters is not the only goal of single-cell methylome sequencing. For the purpose of cell type clustering, sequencing the transcriptome or chromatin accessibility are much easier. The real utility for single-cell methylome sequencing is to (i) derive cell-type specific full methylome; (ii) study cell-to-cell variability of the DNA methylome. It would be really important for the authors to discuss the readiness of sciMETv2.LA and sciMETv2.SL for these purposes. At the level of efficiency and coverage they determined, what is the amount of effort and estimated budget required to achieve the two goals above, for a typical tissue type? What is feasible now, and what remains challenging? Is there any specific niche(s) for sciMETv2.LA vs sciMETv2.SL?

Reviewer #2 (Remarks to the Author):

Single-cell epigenome analysis is a fast-growing field that has attracted enormous attention. Among the cutting-edge technologies, sci-MET is certainly very powerful in that it is capable of generating high-throughput single-cell methylomes. In this manuscript, the authors described sciMETv.2, which enables high-throughput and high-coverage single cell methylome analysis. They introduced two complementary sciMETv.2 methods to achieve high coverage and time&cost-saving individually. It is achieved based on optimization to library preparation, making the method very appealing. The authors employed sciMETv.2 on primary human cortex and verified its ability to identify distinct cell types. This improved version provides sufficient information for assessing complex tissues and is a good fit for Nature Communications if the authors can address some of questions raised below, either through analysis or careful discussions. Below are my concerns.

Major points:

1. First of all, numbering of figures is not appropriately marked in the text. The authors should make a well-organized manuscript after this revision.
2. sciMETv.2 achieved a 14-fold coverage than original version, which equals to ~10% genome. It's an impressive genome coverage for high-throughput single-cell epigenome analysis. The question is when performing cell type discrimination, is there any major difference using sciMET or sciMETv.2? Are there any other application scenarios in need of high-coverage achieved by sci-METv.2?
3. The authors introduced sciMETv2.SL to achieve cost-saving and time-saving, however, there is no comparison results of time or cost consuming between sciMETv2.SL and sciMETv2.LA. And the two complementary version.2 methods need more discussion to be clearly defined in their application scenarios.

Minor points:

4. The authors claimed there are multiple safe stopping points in the workflows, these stopping points should be clearly marked in a detailed protocol which will be quickly adapted by many different labs.
5. The collision rate of sciMETv2.LA seems to be higher than original version, is there some reasonable explanation to this?

Chengqi Yi, Ph.D

Professor, School of Life Sciences

Peking University, Beijing, China.

Reviewer #3 (Remarks to the Author):

Most existing single-cell methylation profiling approaches are plate-based and are difficult to scale up without liquid handling robotics. The sci-MET method published in 2018 was a promising approach to enable combinatorial indexing-based library preparation of single-cell methylome libraries and can be performed without specialized equipment. However, as described in this updated manuscript, sci-MET had several technical issues that reduced the method's robustness. sci-MET v2 was designed to improve the per-cell coverage and library quality, as well as remove the requirement of the custom sequencing primer. Overall, sciMETv2 represents a significant improvement from sciMET that can aid the adoption of the single-cell methylation profiling technique. The manuscript preparation was somewhat sloppy, especially with regard to the citation of figure panels. Although the manuscript has five main figures, the main text only cited Fig. 1-2. This appears to be a major oversight during the preparation of the manuscript.

1. The improved nucleosome disruption using a protocol based on s3-WGS and s3-GCC is a critical component of sciMETv2. However, this improvement was presented with little explanation or data to support the claim. In sci-MET, nuclei were treated with 0.3% SDS and incubated at 42 °C with vigorous shaking for 30 min. In nucleosome disruption protocol in sci-METv2 was not disclosed (only referred to as proprietary ScaleBio reagents), but in s3-WGS protocol, nucleosomes was disrupted with 0.05% SDS at 37 °C for 20 min. The authors should explain why a 6-fold reduction of SDS concentration performed better. The authors should also quantify the uniformity of genomic coverage to show the improved nucleosome disruption protocol indeed outperforms that used by sciMET.

2. Why was a skewed instead of an equal mix of mouse and human nuclei used for the assessment of the doublet rate?

3. Could the authors explain how the doublet rate should be interpreted? On line 70, the dataset only contained 128 mouse cells but 25 mixed cells? Does this result indicate a very high doublet rate, around 20%?

4. Line 93-94. The comparison between sciMET LA and sciMET SL was not fair as the amount of raw reads for sciMET LA was 7.5M whereas only 3.8M for sciMET SL. The authors should down-sample sci-MET LA dataset to make the two datasets more comparable.

Reviewer Response

We thank the reviewers for their generally positive comments and helpful suggestions for improving the manuscript. We believe that the revised version is greatly improved based the added analyses and clarifications.

Reviewer #1 (Remarks to the Author):

In this manuscript the authors reported sciMETv2, a single-cell methylome sequencing method that has significant improvements in terms of the technical performance over the previous version (sciMET) developed by the same research team. With sciMETv2, the authors demonstrated a 15x improvement of genome coverage at the single cell level, by increasing the efficiency of chromatin disruption (prior to tagmentation) and the efficiency of constructing sequencing libraries reusing some technical innovations reported in their recently published S3 protocol. To simplified the experimental workflow, they also presented the data from two version of a shorter ligation-based protocol, both look promising. By applying these protocols to nuclei from human and mouse brains, they benchmarked the technical performance, and compared the results with the published SnnC-seq2 data, which are considered the highest quality data in the field currently. Overall, this study brings the community closer to being able to perform large-scale single-cell methylome routinely. The sciMETv2 method itself, as well as several technical innovations behind this method, would be of high interests to the community of single-cell genomics. I think this manuscript is appropriate for Nature Communications.

We thank the reviewer for the positive comments, and appreciate the comments below which we believe have greatly improved the manuscript.

There are a few areas whether this manuscript can be further improved, mostly from the perspective of potential users.

(1) It would be really helpful to provide an overview of the experimental workflow as a supplementary figure, including the amount of time/effort for each step in each work day, so that the potential users can decide whether this is a method feasible to implement. With that figure, the readers might be able to appreciate better the advantage of sciMETv2.SL over sciMETv2.LA, since based on the performance metrics alone, it's hard to justify sciMETv2.SL.

This is an excellent suggestion. We now include both a flowchart as Supplementary File 2 and include additional details on stopping points in the protocol document, provided as the Supplementary File 3 which includes a detailed step-by-step protocol. The other major factor is cost, with SL roughly 10-fold cheaper. The LA method is more expensive than the original sciMETv1 due to the discontinuation of an ultra-inexpensive klenow fragment reagent that is no longer available, increasing costs for the enzyme by 8-fold and motivating the SL method development. We now include a detailed cost breakdown that is also present in Supplementary File 2 as well as the text and Fig 3c shown below in response to the next comment.

(2) While the authors have demonstrated cell type clustering using both mCH and mCG signatures, and also integrated sciMETv2 data with SnnC-seq2 data, grouping single cells into discrete cell-type specific clusters is not the only goal of single-cell methylome sequencing. For the purpose of cell type clustering, sequencing the transcriptome or chromatin accessibility are much easier. The real utility for single-cell methylome sequencing is to (i) derive cell-type specific full methylome; (ii) study cell-to-cell variability of the DNA methylome. It would be really important for the authors to discuss the readiness of sciMETv2.LA and sciMETv2.SL for these purposes. At the level of efficiency and coverage they determined, what is the amount of effort and estimated budget require to achieve the two goals above, for a typical tissue type? What is feasible now, and what remains challenging? Is there any specific niche(s) for sciMETv2.LA vs sciMETv2.SL?

These are all excellent points. We have now included several additional analyses that directly address these questions and we feel strengthens the manuscript greatly.

The first is addressing the full cell-type specific methylome as well as a comparison of the methods to achieve this goal. Given that any tissue is going to have varied cell type abundances, we elected to carry out a cumulative coverage analysis by adding cells and reporting the methylome coverage (across 100 sampling iterations up to a total of 250 cells). This allows a potential user to determine the approximate number of cells required for a target cell type to achieve a desired level of coverage and design an experiment accordingly (new Figure 3c). We also down sampled the raw reads for the sciMETv2.LA prep to comparable per-cell raw read counts for the sciMETv2.SL preps with the N7 splint and the H10 splint and carried out the same analysis, with a more detailed comparison showing the range of expected coverage across iterations for the coverage-

matched sciMETv2.LA and sciMETv2.SL H10 preps. Indeed, the LA prep performs better when at comparable raw read depth; however, the reduce preparation costs and time may make the SL approach appealing. Ultimately, we believe a heavily optimized version of the SL method may be preferable for all applications, the presentation of the technique here is an initial description that we hope to continue to build upon.

Fig 3c. Cumulative coverage from sampled cells for each method over 100 iterations. Mean values for iterations (left) are shown and include two down sampled variants of the sciMETv2.LA dataset to match the raw read count of the sciMETv2.SL datasets using the H10 (LA, SL-H10 ds.) or N7 (LA, SL-N7 ds.) splints. The distribution of coverage across iterations is shown for the LA, SL-H10 ds. and the SL-H10 datasets (right).

Below is the excerpt from the main text detailing this comparison:

“As expected, the sciMETv2.LA method achieved the highest methylome coverage with the fewest cells, requiring an average of 21 cells to achieve 80% CG-methylome coverage across iterations. However, each cell from the sciMETv2.LA preparation had far greater raw reads sequenced per cell (Fig. 2a,b). We therefore randomly down-sampled the raw read counts to be comparable to the sciMETv2.SL preparations using the N7 (worst-performing) and H10 (best-performing) splints. The LA down-sampled data still produced the greatest coverage per-cell, achieving 80% coverage at an average of 38 cells versus an average of 84 cells for the SL prep using the H10 splint, equating to a 2.2-fold improved efficiency using the LA method over SL at a comparable read count. This difference is considerable; however, the preparation time and costs are greater per-cell for the LA method at 10 hours versus just over 2, and ~\$2,300 per plate versus ~\$230. This equates to a 10-fold cost reduction per cell in preparation costs meaning smaller experiments may achieve a comparable cost point depending on sequencing costs (Supplementary Files 2 & 3; Note: the LA preparation is more expensive than the original sciMETv1 workflow due to increased costs for Klenow fragment).”

And in the Discussion:

“The sciMETv2.SL variant produces less coverage per-cell than the sciMETv2.LA variant at comparable sequencing coverage; however, it is substantially cheaper and faster, making it more favorable as sequencing costs decrease. Furthermore, the simplicity of the splint ligation workflow makes it more appealing for large numbers of cells motivating further optimization to achieve a comparable coverage per cell as with the linear amplification variant.”

The second addition is a new similarity analysis based on the metrics detailed in Hui et. al. Stem Cell Reports 2018 (described as a dissimilarity score). We calculate this metric in an all-by-all fashion and then compare the within-cluster similarities and between cluster similarities specifically at promoter regions (as intergenic and gene body regions are far less variable). This is included as a new Figure 5a.

Finally, we perform a motif-based analysis to assess methylation levels in CG and CH contexts in a cell-type specific manner. We first profiled CTCF which showed little cell type variation as expected, and then for the RE1 binding sites of REST, a protein associated with the repression of neuronal genes and is known to be methylation-sensitive. This revealed clear reduced levels of CG methylation at the binding site specifically in non-neuronal cell types. This analysis is highlighted in the new Figure 5b.

Figure 5 | Cell-cell variability and motif analysis of sciMETv2 data. a. Cell-cell variability as measured by all-by-all methylation similarity at promoter CG sites within each cluster (top), and the mean similarity for cells of different cell types represented as a heatmap (bottom). **b.** Assessment of cell-type-specific methylation patterns for CG (left) and CH (right) centered on motifs for CTCF (top) and REST (RE1, bottom). Reduced CG methylation at RE1 sites in glial cell populations is noted.

Reviewer #2 (Remarks to the Author):

Single-cell epigenome analysis is a fast-growing field that has attracted enormous attention. Among the cutting-edge technologies, sci-MET is certainly very powerful in that it is capable of generating high-throughput single-cell methylomes. In this manuscript, the authors described sciMETv.2, which enables high-throughput and high-coverage single cell methylome analysis. They introduced two complementary sciMETv.2 methods to achieve high coverage and time&cost-saving individually. It is achieved based on optimization to library preparation, making the method very appealing. The authors employed sciMETv.2 on primary human cortex and verified its ability to identify distinct cell types. This improved version provides sufficient information for assessing complex tissues and is a good fit for Nature Communications if the authors can address some of questions raised below, either through analysis or careful discussions. Below are my concerns.

We thank the reviewer for the positive comments and the helpful suggestions that we believe have improved the manuscript substantially.

Major points:

1. First of all, numbering of figures is not appropriately marked in the text. The authors should make a well-organized manuscript after this revision.

This was a version syncing issue and we have ensured that the revised version has properly labeled figures.

2. sciMETv.2 achieved a 14-fold coverage than original version, which equals to ~10% genome. It's an impressive genome coverage for high-throughput single-cell epigenome analysis. The question is when performing cell type discrimination, is there any major difference using sciMET or sciMETv.2? Are there any other application scenarios in need of high-coverage achieved by sci-METv.2?

The major advantage of sciMETv2 is the improved possible coverage over sciMETv1, meaning libraries require more reads to reach saturation, thus ultimately saving costs on sequencing. For an equivalent number of raw reads sciMETv2 will produce more unique, usable reads and higher coverage, thus reducing overall sequencing costs. The other major advantage is the ease of use and robustness of sciMETv2 over v1.

3. The authors introduced sciMETv2.SL to achieve cost-saving and time-saving, however, there is no comparison results of time or cost consuming between sciMETv2.SL and sciMETv2.LA. And the two complementary version.2 methods need more discussion to be clearly defined in their application scenarios.

We now include a detailed breakdown on the cost and time differences between the two methods. sciMETv2.SL has a substantial reduction in both costs and hands-on time; though at the expense of reduced complexity. Ultimately, we believe the SL approach will be preferred, though we expect substantial optimization will need to be carried out to achieve a closer level of coverage compared to that of the LA method. We believe that our new cumulative coverage analysis (Fig. 3c) and detailed cost breakdown (Supplementary File 2) address this, providing possible users with all that is needed to consider each technique. We also view the SL version as earlier in development where it may benefit from further optimization and ultimately become the preferred technique.

Fig 3c. Cumulative coverage from sampled cells for each method over 100 iterations. Mean values for iterations (left) are shown and include two down sampled variants of the sciMETv2.LA dataset to match the raw read count of the sciMETv2.SL datasets using the H10 (LA, SL-H10 ds.) or N7 (LA, SL-N7 ds.) splints. The distribution of coverage across iterations is shown for the LA, SL-H10 ds. and the SL-H10 datasets (right).

“As expected, the sciMETv2.LA method achieved the highest methylome coverage with the fewest cells, requiring an average of 21 cells to achieve 80% CG-methylome coverage across iterations. However, each cell from the sciMETv2.LA preparation had far greater raw reads sequenced per cell (Fig. 2a,b). We therefore randomly down-sampled the raw read counts to be comparable to the sciMETv2.SL preparations using the N7 (worst-performing) and H10 (best-performing) splints. The LA down-sampled data still produced the greatest coverage per-cell, achieving 80% coverage at an average of 38 cells versus an average of 84 cells for the SL prep using the H10 splint, equating to a 2.2-fold improved efficiency using the LA method over SL at a comparable read count. This difference is considerable; however, the preparation time and costs are greater per-cell for the LA method at 10 hours versus just over 2, and ~\$2,300 per plate versus ~\$230. This equates to a 10-fold cost reduction per cell in preparation costs meaning smaller experiments may achieve a comparable cost point depending on sequencing costs (Supplementary Files 2 & 3; Note: the LA preparation is more expensive than the original sciMETv1 workflow due to increased costs for Klenow fragment).”

And in the Discussion:

“The sciMETv2.SL variant produces less coverage per-cell than the sciMETv2.LA variant at comparable sequencing coverage; however, it is substantially cheaper and faster, making it more favorable as sequencing costs decrease. Furthermore, the simplicity of the splint ligation workflow makes it more appealing for large numbers of cells motivating further optimization to achieve a comparable coverage per cell as with the linear amplification variant.”

Minor points:

4. The authors claimed there are multiple safe stopping points in the workflows, these stopping points should be clearly marked in a detailed protocol which will be quickly adapted by many different labs.

This is an excellent suggestion which we now include. This is detailed in the new Supplementary Files 2 (workflow and costs) and 3 (protocol).

5. The collision rate of sciMETv2.LA seems to be higher than original version, is there some reasonable explanation to this?

This is correct. In the original version we sorted an excess of 'events' into each well and found that of the events that were sorted, not all were true cells. In the new sciMETv2 the nuclei that are sorted produce a much cleaner population, yet we still carried through with the elevated number of sorted events per well. This resulted in a higher number of nuclei in each well, due to each event representing an actual nucleus and therefore a higher collision rate.

We now expand on this in the main text, and note that the collision rate follows the expected rate based on number of sorted events, indicating it is tunable to the desired level of cell collision tolerance. We also now provide a recommended number of nuclei to achieve the desired low collision rate. Additionally, we note that the other key component of the mixing experiment is to assess ambient contamination which was very low.

“Notably this results in a higher doublet rate than the original sciMET workflow (19.5% vs <10%). This is due to depositing 22 events per well for downstream processing for which only half were actual viable cells in the original workflow; whereas we achieved a far greater viability with sciMETv2, with nearly all events producing cell profiles. However, the doublet rate was very near the expected count if 100% of events were viable (22.9%), suggesting that the doublet rate is tunable and can be proportionately reduced by decreasing the deposited cells per well. The second component we assessed from the mixed-species experiment was the presence of cell-cell crosstalk which was determined to be <1% across cells for each respective species.”

Chengqi Yi, Ph.D
Professor, School of Life Sciences
Peking University, Beijing, China.

Thank you for your helpful comments on our manuscript!

Reviewer #3 (Remarks to the Author):

Most existing single-cell methylation profiling approaches are plate-based and are difficult to scale up without liquid handling robotics. The sci-MET method published in 2018 was a promising approach to enable combinatorial indexing-based library preparation of single-cell methylome libraries and can be performed without specialized equipment. However, as described in this updated manuscript, sci-MET had several technical issues that reduced the method's robustness. sci-MET v2 was designed to improve the per-cell coverage and library quality, as well as remove the requirement of the custom sequencing primer. Overall, sciMETv2 represents a significant improvement from sciMET that can aid the adoption of the single-cell methylation profiling technique. The manuscript preparation was somewhat sloppy, especially with regard to the citation of figure panels. Although the manuscript has five main figures, the main text only cited Fig. 1-2. This appears to be a major oversight during the preparation of the manuscript.

We thank the reviewer for the generally positive comments and their suggestions to further improve the manuscript. The figure panel labeling was a version syncing issue and we have ensured that the revised version has properly labeled figures.

1. The improved nucleosome disruption using a protocol based on s3-WGS and s3-GCC is a critical component of sciMETv2. However, this improvement was presented with little explanation or data to support the claim. In sci-MET, nuclei were treated with 0.3% SDS and incubated at 42 °C with vigorous shaking for 30 min. In nucleosome disruption protocol in sci-METv2 was not disclosed (only referred to as proprietary ScaleBio reagents), but in s3-WGS protocol, nucleosomes was disrupted with 0.05% SDS at 37 °C for 20 min. The authors should explain why a 6-fold reduction of SDS concentration performed better. The authors should also quantify the uniformity of genomic coverage to show the improved nucleosome disruption protocol indeed outperforms that used by sciMET.

The reviewer raises a good point. We previously observed improved coverage uniformity for the s3-WGS assay compared to our original sci-DNA-seq technique and assumed that coverage uniformity would also be improved for the sciMET assay. However, this is not the case, the uniformity of the sciMETv1 was already

ideal, as is the uniformity of the new sciMETv2 conditions. We assess this based on TSS enrichment using a calculation that produces values centered around 1 for bulk shotgun WGS data (many TSS enrichment calculations, such as the ENCODE version, produces a value close to 3 for bulk shotgun WGS data). We believe TSS enrichment is the best assessment due to the direct goal of ablating the chromatin accessibility signal that is present when tagging intact nuclei.

We now include this comparison alongside an example sci-ATAC preparation and original sciMETv1 data and alter the text to state that it achieves higher read counts with comparable coverage uniformity to the original sciMETv1 workflow (Fig. 1b, left). We also expand on the improvement by detailing that the reduction in formaldehyde concentration allows for sufficient crosslinks to maintain nuclei integrity without inhibiting tagmentation which we believe was a major driver of the reduced complexity of previous preparations. The reduction in SDS concentration is designed to retain nuclei integrity with reduced fixation yet still high enough to achieve nucleosome disruption.

Fig 1b. Nucleosome disruption effectiveness as measured by raw transcription start site (TSS) enrichment. Tagmentation of intact nuclei preserves TSS enrichment which is ablated by nucleosome disruption in sciMET assays.

“The second major improvement shared between the techniques is the use of optimized nucleosome disruption methods that we previously described for s3-WGS and s3-GCC for single-cell genome sequencing or genome sequencing plus chromosome conformation, respectively¹². These improvements include a reduction in formaldehyde concentration, thus reducing the amount of fixation which may impede tagmentation efficiency while achieving enough fixation to preserve nuclei integrity during the detergent (SDS) based nucleosome disruption, which was also reduced. Together these optimizations achieve greater tagmentation efficiency, which translates to increased per-cell coverage, with a comparable coverage uniformity as assessed by the ablation of transcription start site (TSS) enrichment (Fig. 1b).”

2. Why was a skewed instead of an equal mix of mouse and human nuclei used for the assessment of the doublet rate?

This was done to save on costs. Experiments that allow for assessment of doublet rate while allowing for the species of interest to be analyzed is preferred over an entire experiment dedicated to the mixed species.

3. Could the authors explain how the doublet rate should be interpreted? On line 70, the dataset only contained 128 mouse cells but 25 mixed cells? Does this result indicate a very high doublet rate, around 20%?

This is correct. In the original version we sorted an excess of ‘events’ into each well and found that of the events that were sorted, not all were true cells. In the new sciMETv2 the nuclei that are sorted produce a much cleaner population, yet we still carried through with the elevated number of sorted events per well. This resulted in a higher number of nuclei in each well, due to each event representing an actual nucleus and therefore a higher collision rate.

We now expand on this in the main text, and note that the collision rate follows the expected rate based on number of sorted events, indicating it is tunable to the desired level of cell collision tolerance. We also now provide a recommended number of nuclei to achieve the desired low collision rate. Additionally, we note that the other key component of the mixing experiment is to assess ambient contamination which was very low.

“Notably this results in a higher doublet rate than the original sciMET workflow (19.5% vs <10%). This is due to depositing 22 events per well for downstream processing for which only half were actual viable cells in the original workflow; whereas we achieved a far greater viability with sciMETv2, with nearly all events producing cell profiles. However, the doublet rate was very near the expected count if 100% of events were viable (22.9%), suggesting that the doublet rate is tunable and can be proportionately reduced by decreasing the deposited cells per well. The second component we assessed from the mixed-species experiment was the presence of cell-cell crosstalk which was determined to be <1% across cells for each respective species.”

4.Line 93-94. The comparison between sciMET LA and sciMET SL was not fair as the amount of raw reads for sciMET LA was 7.5M whereas only 3.8M for sciMET SL. The authors should down-sample sci-MET LA dataset to make the two datasets more comparable.

We thank the reviewer for the suggestion and now downsample to a comparable read count and report relevant statistics in the text along with a new analysis detailing cumulative CG coverage by cell number, provided as a new Figure 3c. We believe that accumulated coverage is the most important metric that is directly relevant to raw read devotion to a library and most important for considering experimental designs.

Fig 3c. Cumulative coverage from sampled cells for each method over 100 iterations. Mean values for iterations (left) are shown and include two down sampled variants of the sciMETv2.LA dataset to match the raw read count of the sciMETv2.SL datasets using the H10 (LA, SL-H10 ds.) or N7 (LA, SL-N7 ds.) splints. The distribution of coverage across iterations is shown for the LA, SL-H10 ds. and the SL-H10 datasets (right).

“As expected, the sciMETv2.LA method achieved the highest methylome coverage with the fewest cells, requiring an average of 21 cells to achieve 80% CG-methylome coverage across iterations. However, each cell from the sciMETv2.LA preparation had far greater raw reads sequenced per cell (Fig. 2a,b). We therefore randomly down-sampled the raw read counts to be comparable to the sciMETv2.SL preparations using the N7 (worst-performing) and H10 (best-performing) splints. The LA down-sampled data still produced the greatest coverage per-cell, achieving 80% coverage at an average of 38 cells versus an average of 84 cells for the SL prep using the H10 splint, equating to a 2.2-fold improved efficiency using the LA method over SL at a comparable read count. This difference is considerable; however, the preparation time and costs are greater per-cell for the LA method at 10 hours versus just over 2, and ~\$2,300 per plate versus ~\$230. This equates to a 10-fold cost reduction per cell in preparation costs meaning smaller experiments may achieve a comparable cost point depending on sequencing costs (Supplementary Files 2 & 3; Note: the LA preparation is more expensive than the original sciMETv1 workflow due to increased costs for Klenow fragment).”

REVIEWERS' COMMENTS

Reviewer #1 (Remarks to the Author):

The authors have fully addressed all my concerns in this revision. It is appropriate to be published on Nature Communications.

Reviewer #2 (Remarks to the Author):

In this revised manuscript, the authors have addressed all the points well. The revised results and discussion may help the readership to be more acquainted with sciMETv2 and its performance. Overall, I would recommend accepting for publication on Nature Communications without further revisions.

Chengqi Yi, Ph.D

Professor, School of Life Sciences

Peking University, Beijing, China.

Reviewer #3 (Remarks to the Author):

The authors have successfully addressed all my concerns, and I found the revised manuscript significantly improved.